# A Multi-Focal Image Fusion Network for Implantation Outcome Prediction of Blastocyst

**Yi Cheng**[1*]                                                                    CHENGY1@ZJU.EDU.CN
**Tingting Chen**[2*]                                                   TRISTA_CHEN0603@ZJU.EDU.CN
**Yaojun Hu**[2*]                                                            YAOJUNHU@ZJU.EDU.CN
**Xiangqian Meng**[3†]                                                  MENGXQ@JXR-FERTILITY.COM
**Zuozhu Liu**[4]                                                      ZUOZHULIU@INTL.ZJU.EDU.CN
**Danny Z. Chen**[5]                                                               DCHEN@ND.EDU
**Jian Wu**[6,7]                                                          WUJIAN2000@ZJU.EDU.CN
**Haochao Ying**[7†]                                                    HAOCHAOYING@ZJU.EDU.CN

[1]*School of Software Technology, Zhejiang University, Ningbo, China*

[2]*College of Computer Science and Technology, Zhejiang University, Hangzhou, China*

[3]*Sichuan Jinxin Xinan Women & Children Hospital, Chengdu, China*

[4]*ZJU-UIUC Institute, Zhejiang University, Haining, China*

[5]*Department of Computer Science and Engineering, University of Notre Dame, Notre Dame, USA*

[6]*Second Affiliated Hospital School of Medicine Zhejiang University, Hangzhou, China*

[7]*School of Public Health, Zhejiang University, Hangzhou, China*

**Editors:** Under Review for MIDL 2024

## Abstract

Accurately predicting implantation outcomes based on blastocyst developmental potential is valuable in in-vitro fertilization (IVF). Clinically, embryologists analyze multiple focal-plane images (FP-images) to comprehensively assess embryo grades, which is extremely cumbersome and easily prone to inconsistency. Developing automatic computer-aided methods for analyzing embryo images is highly desirable. However, effectively fusing multiple FP-images for prediction remains a largely under-explored issue. To this end, we propose a novel Multiple Focal-plane Image Fusion Network, called MFIF-Net, to predict implantation outcomes of blastocyst. Specifically, our MFIF-Net consists of two subnetworks: a Core Image Generation Network (CI-Gen) and a Key Feature Fusion Network (KFFNet). In CI-Gen, we fuse multiple FP-images to generate a *core image* by pixel-wise weighting since different FP-images can have different focus positions. To further capture key features in each FP-image, we propose KFFNet to extract key information from the FP-images again and fuse them with the core image. In KFFNet, a Fusion Module is designed to capture key information of each FP-image, for which Squeeze Multi-Headed Attention is developed to exchange features and mitigate computationally intensive issue in attention. Comprehensive experiments validate the superiority and the rationality of our MFIF-Net approach over state-of-the-art methods in various metrics. Ablation studies also confirm the positive impact of each component in our MFIF-Net. The code will be publicly available upon acceptance.

**Keywords:** Blastocyst implantation prediction, in-vitro fertilization, multi-modalities

---

[*] Contributed equally

[†] Corresponding authors

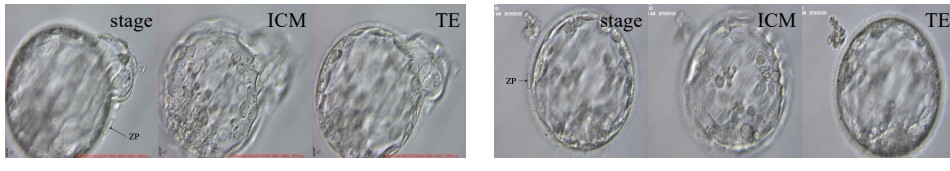

$(a)$ positive example $\qquad$ $(b)$ negative example

Figure 1: Examples of microscopic images at different focal planes of blastocysts.

## 1. Introduction

In vitro fertilization (IVF) is the most prevalent treatment for infertility. Due to the inherent risks of multiple pregnancies (Fanelli et al., 2012), it is critical to select high quality embryo for single-embryo transfer, to produce one healthy baby. Typically, Embryo transfer (ET) involves cleavage stage ET and blastocyst stage ET. According to the recent finding (Papanikolaou et al., 2005), the blastocyst stage ET significantly enhances implantation rates. Thus, in clinical practice, embryologists often manually analyze multiple blastocyst stage embryo images to identify those with the highest likelihood of successful implantation. However, this manual analysis is laborious and subject to considerable variability (Sundvall et al., 2013; Storr et al., 2017). To help embryologists effectively evaluate blastocyst quality and accurately predict implantation outcomes, it is highly desirable to develop automatic computer-aided methods for analyzing embryo images.

Recent researches in computer-aided diagnosis (CAD) for embryo analysis mainly focus on three key tasks: stage classification (Khan et al., 2016; Lukyanenko et al., 2021; Lockhart et al., 2021), blastocyst segmentation, (Harun et al., 2019; Rad et al., 2020) and blastocyst grading (Khosravi et al., 2019). While stage classification and blastocyst segmentation are crucial preliminary steps in embryo analysis, they do not directly predict implantation outcomes. Current blastocyst grading methods (Khosravi et al., 2019) evaluated implantation rates by categorizing a single microscopic image into various grades. However, this approach struggles to accurately represent the three-dimensional nature of embryos, particularly the inner cell mass (ICM) and trophectoderm (TE), in a single image. Clinically, embryologists evaluate the stage, inner cell mass (ICM), and trophectoderm (TE) of a blastocyst independently to derive a comprehensive score indicative of its transfer potential. The stage is determined by the blastocyst's developmental stage and its interaction with the zona pellucida (ZP), while ICM and TE refer to specific cellular components of the blastocyst. As depicted in Fig. 1, 'stage' images show the blastocyst's breakthrough of the ZP while 'ICM' and 'TE' images highlight specific areas of the blastocyst. However, capturing these features distinctly in a single image is challenging. Therefore, developing an image-fusion technique for accurate prediction of blastocyst implantation outcomes is imperative.

Currently, joint analysis of multiple focal-plane (FP) images of embryos is still in its infancy. Zeman et al. (Zeman et al., 2021) chose three FP-images and concatenated them directly to predict embryo quality, treating the three FP-images as equally important. However, embryonic information contained in different FP-images is different, and treating them as equally important may make it difficult to fully exploit the features captured by different focal planes. Worse, known multi-modal fusion methods, no matter early-, mid-, late-, and hybrid-fusion types (Zeman et al., 2021; Nagrani et al., 2021; Pang et al., 2020; Zhou et al., 2020), neglect extraction of the specific information or key information (e.g., ICM

area in Fig. 1($b$) of each modality), which may have strong correlation with the final result. Moreover, most known fusion methods utilize two modalities, which are relatively easy to fuse. However, the challenge in predicting blastocyst implantation outcomes involves the analysis of three FP images with different key information, necessitating the development of more effective multi-modal fusion techniques.

To this end, we propose a novel Multiple Focal-plane Image Fusion Network (MFIF-Net), which utilizes three FP-images of a blastocyst as input and predicts implantation outcomes. Specifically, MFIF-Net consists of two sub-networks: the Core Image Generator (CI-Gen) and the Key Feature Fusion Network (KFFNet). In CI-Gen, since the three FP-images focus on different positions, we first fuse the three FP-images to generate a 'clear' *core image* by pixel-wise weighting. However, information loss will occur in the core image generation process since there are overlaps among the three FP-images. Therefore, in KFFNet, to further utilize key information in each FP-image, we propose a Fusion Layer to capture key features by a Fusion Module in each focal plane, and fuse them with the core image features. Note that in the Fusion Module, we apply spatial-channel separated Squeeze Multi-Headed Attention (SMHA) blocks for efficient information exchange and feature enhancement. In summary, we achieve feature fusion of three focal-plane images at each stage through the core image and Fusion Module, effectively reducing redundancy and better integrating essential information.

**Contributions.** 1) We propose a novel Multiple Focal-plane Image Fusion Network for implantation outcome prediction of blastocyst. This network uniquely integrates key information from the multiple FP-image fusion perspective, which is under-explored in prior work. 2) We design a new plug-and-play feature interaction block tailored for facilitating information exchange and mitigating computational intensity in attention mechanisms, to address the limitation of current methods in failing to extract key information from various locations in FP images. 3) We conduct extensive experiments to demonstrate the superior performance of our MFIF-Net over state-of-the-art methods in various metrics, and validate the rationality of each component in MFIF-Net through sufficient ablation studies.

## 2. Methodology

As illustrated in Fig. 2, we propose MFIF-Net for analyzing multiple FP-images of the blastocyst to predict implantation outcomes. Specifically, MFIF-Net executes two main steps to perform multi-FP-image fusion and utilizes the specific features of each FP-image. In the first step, given that different FP-images have varying focus points and significance in blastocyst assessment, we generate a core image through weighted fusion of these images. However, the initial fusion in the core image can result in information loss due to overlapping focus areas and insufficient information fusion. Thus, in the second step, the designed KFFNet module further exploits the importance of each FP-image and integrates it with the core image to enhance feature learning. Below we elaborate our MFIF-Net in detail.

### 2.1. Core Image Generator (CI-Gen)

In common modal fusion methods, two modalities are usually fused with each other, but this fusion strategy is not suitable for fusing three modalities. This is because the feature extraction layer of each modality can cover key information of the other modalities, which

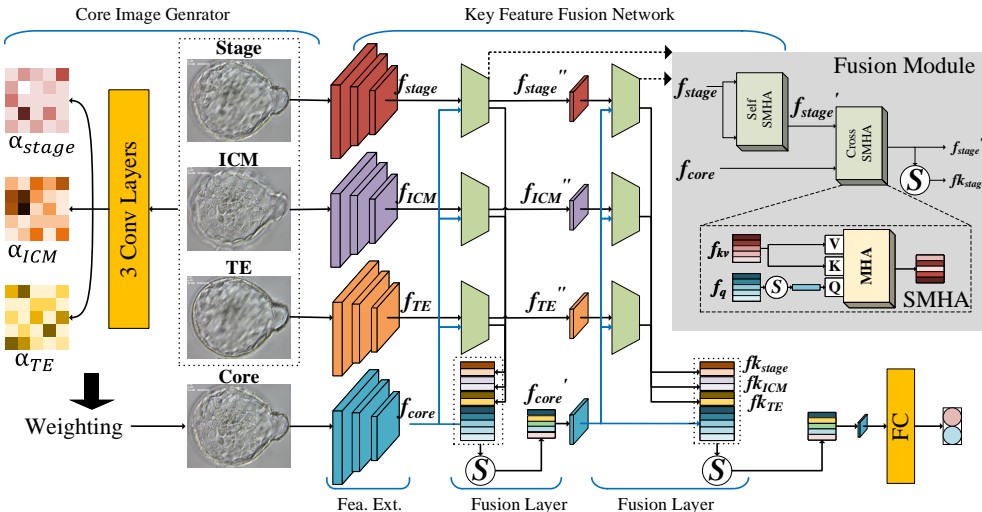

Figure 2: An overview of MFIF-Net. ⓈS denotes a channel-reduced convolutional layer or an average pooling layer. A dotted rectangle indicates concatenation.

will cause the feature fusion to be ineffective. We verify this observation through early fusion and late fusion in our comparative experiment. Hence, we propose the CI-Gen sub-network for fusing three modalities. We first perform a preliminary fusion of the three FP-images by generating a core image. Since different FP-images (see Fig. 1) focus on different regions of blastocyst, we seek to produce a focus-on-everywhere image by combining every focused area of each FP-image and considering their relative importance. Thus, as shown in the left part of Fig. 2, three 'RGB' FP-images ($I_{stage}$, $I_{ICM}$, and $I_{TE}$) are concatenated to form a 9-channel tensor as input. After going through three convolutional layers (expressed by the cubic operation in Eq. (1)), the output is a 3-channel tensor $\alpha$ (composed of $\alpha_{stage}$, $\alpha_{ICM}$, and $\alpha_{TE}$), which indicates a weight map for each FP-image. Finally, the core image $I_{core}$ is generated by weighted summation of the three FP-images and their corresponding predicted weights in $\alpha$, as follows:

$$\alpha = [\alpha_{stage}, \alpha_{ICM}, \alpha_{TE}] = Conv2d(Concat(I_{stage}, I_{ICM}, I_{TE}))^3, \tag{1}$$

$$I_{core} = \sum \alpha_y * I_y, \ y \in \{stage, ICM, TE\}. \tag{2}$$

### 2.2. Key Feature Fusion Network (KFFNet)

After generating the core image $I_{core}$, we apply KFFNet to the four images ($I_{core}$, $I_{stage}$, $I_{ICM}$, and $I_{TE}$) for further feature extraction and fusion. First, the feature extraction layers generate three focal plane feature maps and a core feature map for these four images. After the third feature extraction layer, we use two Fusion Layers to capture key features in these focal plane feature maps and fuse them with the core feature map. Finally, a fully-connected layer predicts implantation outcomes from the output of KFFNet.

**Feature Extraction.** We take four individual ResNet-18's (He et al., 2016) as the feature extraction modules for the three FP-images and the core image, all of which use

ImageNet (Deng et al., 2009) pre-trained weights. After three feature extraction layers, the feature maps of these four images are $f_{stage}$, $f_{ICM}$, $f_{TE}$, and $f_{core}$, respectively.

**Fusion Layer.** We devise the Fusion Layer to capture and fuse key features in the focal plane feature maps. Since the three focal plane feature maps are processed in the same way, we describe only the fusion process for the stage focal plane feature map $f_{stage}$. First, we utilize the Fusion Module (as described below) to enhance $f_{stage}$ and extract key features $fk_{stage}$ for further feature fusion promotion. After that, the Fusion Layer concatenates key features $fk$ of each focal plane with $f_{core}$, and the concatenated features are re-fused by a channel-reduced convolutional layer for further fusion:

$$f_{concat} = Concat(fk_{stage}, fk_{ICM}, fk_{TE}, f_{core}), \tag{3}$$

$$f'_{core} = Conv(f_{concat}). \tag{4}$$

**Fusion Module.** The Fusion Module is applied between each focal plane feature map and the core feature map. The top-right area of Fig. 2 shows the processing pipeline, which takes core features $f_{core}$ and stage focal plane features $f_{stage}$ (use stage as example) as input. SMHAs undertake the function of information exchange and feature enhancement inside the Fusion Module, as follows. First, self-SMHA enhances features in $f_{stage}$ and generates $f'_{stage}$. After that, information exchange is conducted by cross-SMHA to produce $f''_{stage}$ using $f_{core}$ and $f'_{stage}$. The above steps complete information interaction and feature enhancement. To avoid information redundancy and retain the most significant information, key features are generated from $f''_{stage}$ by a channel-reduced convolutional layer :

$$f'_{stage} = self-SMHA(f_{stage}, f_{stage}), \tag{5}$$

$$f''_{stage} = cross-SMHA(f_{core}, f'_{stage}), \tag{6}$$

$$fk_{stage} = Conv(f''_{stage}). \tag{7}$$

**SMHA.** Inspired by TransFuser (Prakash et al., 2021), we develop a new plug-and-play feature interaction block, called SMHA block. In TransFuser, MHA (Vaswani et al., 2017) abandons the traditional CNN method of extracting features from 3D tensors through convolution kernels, and instead computes the similarity between 2D tensors, query $f_x$ and key $f_y$, of length $dk$. Then, the result of similarity is multiplied with the values in $f_y$, as:

$$MHA(f_x, f_y) = Softmax(\frac{f_x W^Q \cdot (f_y W^K)^T}{\sqrt{dk}}) \cdot (f_y W^V), \tag{8}$$

where $W^Q \in \mathbb{R}^{dk \times dk}$, $W^K \in \mathbb{R}^{dk \times dk}$, and $W^V \in \mathbb{R}^{dk \times dk}$ are query, key, and value projection matrices, respectively.

In order to exchange information between CNN features by MHA, we reshape the CNN features from 3D to 2D to satisfy the input form of MHA. However, the flattened features reach sizes of $196 \times 256$ and $49 \times 512$ (take the output of the last two layers of ResNet-18 as examples), which will greatly increase the amount of computation for the network. Meanwhile, inspired by P3D (Qiu et al., 2017), dimension-separated feature extraction leads to better performance. For these two reasons, we design SMHA to improve MHA by squeezing the spatial or channel dimension of the query feature map, as follows.

(1) Spatial SMHA: The query features and key-value features are $f_q \in \mathbb{R}^{C \times H \times W}$ and $f_{kv} \in \mathbb{R}^{C \times H \times W}$. In spatial-SMHA, $f_q$ is transformed into $\mathbb{R}^{1 \times C}$ by an Average-Pooling layer, $f_{kv}$ is reshaped to $\mathbb{R}^{(H \times W) \times C}$, and $dk$ in MHA is equal to the channel number. Spatial SMHA can be described as:

$$Spatial - SMHA(f_q, f_{kv}) = MHA(AvgPool(f_q), f_{kv}). \qquad (9)$$

(2) Channel SMHA: Similarly, $f_q$ goes through a convolutional layer, and the number of channels is reduced to a single channel as $\mathbb{R}^{1 \times H \times W}$. $f_{kv}$ is reshaped to $\mathbb{R}^{C \times (H \times W)}$, and $dk$ in MHA is equal to $H \times W$. Channel-SMHA can be specified as:

$$Channel - SMHA(f_q, f_{kv}) = MHA(Conv(f_q), f_{kv}). \qquad (10)$$

In self-SMHA, $f_q$ and $f_{kv}$ are both focal plane feature maps, while in cross-SMHA, $f_q$ is the core feature map. We give both performance comparisons and computation costs of different SMHA combinations in the experiments and appendix, respectively.

## 3. Experimental Results

The dataset comprises microscopic images of 643 human embryos, sourced from a collaborating hospital and ethically approved, divided into two categories based on post-surgery results: successful implantation (n=310) and implantation failure (n=333). For each embryo, we manually take three microscopic images of different focal planes: stage, ICM, and TE. Due to the inherent movement of embryos during imaging, the stage FP-image was designated as a reference for aligning the other images. We evaluate the performance of our MFIF-Net using accuracy (ACC, %), sensitivity (SEN, %), positive predictive value (PPV, %), negative predictive value (NPV, %), F1 score, and area under the receiver operating characteristic curve (AUC) compared to previous methods. To enhance the robustness of our findings and avoid biases from a limited dataset, we adopt a stratified sampling method, culminating in a five-fold cross-validation approach. The results presented are the aggregated averages from this comprehensive cross-validation process.

### 3.1. Comparison to State-of-the-Art Methods

We modify known state-of-the-art (SOTA) methods to fit our dataset. (1) Erlich et al. (Erlich et al., 2022) used ResNet50 (He et al., 2016) as the feature extractor. (2) STEM (Liao et al., 2021) classified blastocyst and nonblastocyst images with DenseNet (Huang et al., 2017). (3) STORK (Khosravi et al., 2019) trained InceptionNet-V1 (Szegedy et al., 2015) for embryo quality grading. (4) Fordham et al. (Fordham et al., 2022) used EfficientNetV2 (Tan and Le, 2021) as the image encoder. These methods cover the widely-used CNN models, and all of them achieved state-of-the-art performance on their respective tasks. Hence, we migrate these methods to test on our dataset and apply early fusion and late fusion on them for fair comparison. Specifically, Early Fusion (Zeman et al., 2021) concatenats the grayscales of the three FP-images into an 'RGB' image, while Late Fusion uses three individual backbones to extract feature maps and concatenates them before the classifier. Each model is retrained on our dataset, and the best parameters for accuracy are selected for testing. As shown in Table 1, compared with the known SOTA methods,

Table 1: Quantitative comparison of MFIF-Net and SOTA methods on five-fold cross-validation. (E) denotes early fusion and (L) indicates late fusion. We use **bold** to indicate the best results and underline to represent the second-best results.

| Method | ACC (%) | F1 | AUC | SEN (%) | PPV (%) | NPV (%) |
|---|---|---|---|---|---|---|
| (E) Erlich et al. | 59.0 | 58.3 | 55.0 | 52.9 | 58.4 | 59.4 |
| (E) STEM | 59.3 | 57.2 | 56.0 | 50.0 | 59.4 | 59.1 |
| (E) STORK | 60.8 | 60.8 | 60.1 | 61.6 | 59.2 | 62.5 |
| (E) Fordham et al. | 58.5 | 55.5 | 55.0 | 56.1 | 57.5 | 59.7 |
| (L) Erlich et al. | 56.9 | 51.8 | 55.5 | 56.5 | 55.2 | 58.2 |
| (L) STEM | 58.3 | 55.5 | 54.8 | 41.6 | 59.8 | 57.3 |
| (L) STORK | 59.1 | 58.2 | 56.2 | 63.9 | 57.0 | 61.7 |
| (L) Fordham et al. | 57.4 | 54.4 | 54.2 | 31.6 | 61.5 | 55.9 |
| MFIF-Net (ours) | **65.6** | **65.6** | **62.8** | **64.5** | **64.5** | **66.7** |

our MFIF-Net outperforms them in all the evaluation metrics. For instance, our accuracy is 4.8% higher than the best existing method, and we achieve a 3% increase in positive predictive value and a 4.2% increase in negative predictive value. This is because CI-Gen initially eliminates redundancy and focuses on the significant regions of each FP-image. The subsequent Fusion Module captures key features of FP-images and fuses them with the core feature map, which further enhances multi-modal fusion. Therefore, our MFIF-Net comprehensively outperforms the Early Fusion and Late Fusion methods.

### 3.2. Ablation Study

We design ablation experiments shown in Table 2, 3 and 4 to verify the improvement brought by each component in our MFIF-Net.

**Effects of Different Types of FP-images and Core Image.** To demonstrate the importance of different types of FP-images, we conduct experiments on single-type FP-image classification, as shown in Table 2. In this table, ICM, TE, and stage represent experiments using only one type of FP-images for classification. "Concat" indicates an experiment where the three types of FP-images are concatenated and used for classification (Zeman et al., 2021), and "Core Image" represents an experiment using only the core image generated by our proposed Core Image Generator. From the results in Table 2, it can be observed that both "Concat" and "Core Image" outperform the models using only a single type of FP-images in all the metrics, indicating that utilizing information from all the three types of images effectively improves the model performance. Furthermore, our proposed Core Image Generator outperforms "Concat" in most the metrics, with only a slight decrease of 0.1 in F1 score, demonstrating that our Core Image Generator achieves better fusion of different FP-image types by simply weighting the three FP-images.

**Effects of Different Modules.** To validate the effectiveness of the two components in our method, CI-Gen and KFFNet, we conduct experiments and the results are shown in Table 3. Here, "Concat" refers to the fusion of the three types of FP-images, which is consistent with the results in Table 2. "Core Image" represents the experiments using only the core image generated by CI-Gen, and "Fusion Layer" denotes the model that combines the three types of FP-images using the proposed fusion layer in KFFNet. From the results in Table 3, it can be observed that the benefits of the Fusion Layer are not as significant as those of the core image. However, considering the information loss in the Core Image version, we add the Fusion Module with the core image and the three FP-images

Table 2: Effects of three different types of FP-images and core image.

| Method | ACC (%) | F1 | AUC | SEN (%) | PPV (%) | NPV (%) |
|---|---|---|---|---|---|---|
| ICM | 57.1 | 52.6 | 55.3 | 48.4 | 56.4 | 57.2 |
| TE | 58.3 | 57.7 | 55.2 | 58.7 | 56.8 | 60.0 |
| Stage | 58.2 | 56.3 | 54.2 | 50.6 | 57.5 | 58.3 |
| Concat (Zeman et al., 2021) | 61.4 | **61.4** | 60.4 | 59.4 | 60.3 | 62.4 |
| Core Image | **62.2** | 61.3 | **60.8** | **62.6** | **60.6** | **63.7** |

Table 3: Effects of different modules.

| Method | ACC (%) | F1 | AUC | SEN (%) | PPV (%) | NPV (%) |
|---|---|---|---|---|---|---|
| Concat (Zeman et al., 2021) | 61.4 | 61.4 | 60.4 | 59.4 | 60.3 | 62.4 |
| Core Image | 62.2 | 61.3 | 60.8 | 62.6 | 60.6 | 63.7 |
| Fusion Layer | 61.8 | 60.1 | 58.7 | 53.2 | 62.1 | 61.3 |
| MFIF-Net | **65.6** | **65.6** | **62.8** | **64.5** | **64.5** | **66.7** |

Table 4: Effects of different combinations of self-SMHA and cross-SMHA.

| Self-SMHA | Cross-SMHA | ACC | F1 | AUC | SEN | PPV | NPV |
|---|---|---|---|---|---|---|---|
| Channel | Channel | 64.1 | 63.8 | 61.3 | **69.0** | 61.5 | **67.1** |
| Spatial | Spatial | 63.8 | 63.4 | 62.3 | 60.0 | 63.3 | 64.2 |
| Channel | Spatial | 64.2 | 64.1 | 62.0 | 56.1 | **65.3** | 63.6 |
| Spatial | Channel | **65.6** | **65.6** | **62.8** | 64.5 | 64.5 | 66.7 |

to supplement information and enhance features. The final results demonstrate that the overall performance of our MFIF-Net significantly outperforms the other versions in Table 3.

**Effects of Different Combinations of Self-SMHA and Cross-SMHA.** To examine the effects brought by different SMHA combinations, we conduct an additional ablation experiment presented in Table 4. Here, the first column and the second column respectively indicate whether the SMHA used in self-SMHA and cross-SMHA is channel-SMHA or spatial-SMHA. As shown in Table 4, the combinations with different SMHAs perform better than the combinations with the same SMHA modules. This is because the Fusion Module made up with the same SMHAs cannot fully enhance features. In addition, the channel-channel model has the best SEN and NPV. This is because this model is weak in spatial feature extraction and cannot identify the targets in the stage, ICM, and TE areas well. Therefore, this model is more likely to classify samples as positive, which leads to an increase of SEN and NPV. The spatial-channel combination is better than the channel-spatial one. We believe this is because the spatial information in blastocyst's FP-images is quite obvious, and self-spatial-SMHA can generate useful feature maps without the core image's information. Then, with the supervision of the core image, the most valuable channels are enhanced for further fusion.

## 4. Conclusions

In this paper, we proposed a novel Multiple Focal-plane Image Fusion Network (MFIF-Net) for implantation outcome prediction of blastocyst. To address the significant limitation of existing methods in extracting key information from different focal plane images, the Core Image Generator innovatively combines key information from multiple focal plane (FP) images at different stages to generate a core image, which is then utilized in the middle and late fusion stages by Squeeze Multi-Headed Attention in Key Feature Fusion Network. Note that our method is scalable for multiple image fusion. Extensive experimental comparisons and detailed ablation studies demonstrate the superior performance of MFIF-Net.

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

## Appendix A. Experiments Setups

We use PyTorch to build and train our MFIF-Net, and use the SGD optimizer with momentum = 0.9, weight decay = $1 \times 10^{-4}$, $\lambda = 1$, and learning rate = $3 \times 10^{-3}$. We train the network for 100 epochs with a mini-batch of 8. We first align the three FP-images because blastocyst often moves slightly when photographing the multiple FP-images. The input images are scaled to size $224 \times 224$. Random cropping, flipping, and rotation are used for data augmentation during training; only center cropping is used in the inference stage. The Fusion Module is applied after the $3^{rd}$ layer, and the squeeze output channel number in the Fusion Module is 4 in our experiments. The three convolutional layers in CI-Gen use $13 \times 13$ convolutional kernel, and their input-output channels are $9 - 64, 64 - 128, 128 - 3$, respectively. Spatial-Channel SMHA combination is used in Fusion Module.

## Appendix B. Additional Baselines Comparsion

The following additional conclusions are based on the analysis of Table 1.

(a) In both the Early Fusion and Late Fusion groups, STORK outperforms known methods across most of the metrics. This can be attributed to the presence of the Inception module within STORK, which incorporates parallel convolutional layers and pooling layers, along with convolutional kernels of varying scales. This design enables the model to capture features in different scales, enhancing its ability to fuse information from various modalities more effectively. As a result, STORK demonstrates an improved capacity for understanding and representing multi-modal data.

(b) The Early Fusion method in each backbone model has better classification performance than the Late Fusion one. We believe this is due to the high similarity among the three FP-images. Similar images bring redundant feature vectors before Late Fusion, which brings many noisy features and results in worse classification performance.

## Appendix C. Computational Cost Comparison

Squeeze Multi-Head Attention (SMHA) replaces the original query with the squeezed one for computational cost reduction. Table 5 reports that SMHA reduces the computational costs of MHA to 50.32%, 65.76%, and 58.04% with channel SMHA, spatial SMHA, and the overall Fusion Module (in Fusion Layer 1), respectively. We can conclude that our SMHA mitigates the computationally expensive problem of transformer in vision tasks.

Table 5: Computational cost comparison between MHA and SMHA.

| Method | MFlops (in Fusion Layer 1) |
|---|---|
| MHA | 157.35 |
| channel-SMHA | 79.18 (50.32%) |
| spatial-SMHA | 103.48 (65.76%) |
| Fusion Module (MHA) | 314.7 |
| Fusion Module (SMHA) | 182.67 (58.04%) |

