# OpenReview forum: "MFIF-Net: A Multi-Focal Image Fusion Network for Implantation Outcome Prediction of Blastocyst"
_MIDL.io/2024/Conference — MIDL 2024 Poster_

### Official Review · Reviewer_eE6q · 2024-02-20

**Confidence:** 3
**Preliminary Rating:** 4
**Recommendation:** Poster
**Final Rating:** 4

**Summary:**

The paper is focused on blastocyst implantation prediction based on microscopy image analysis. In particular, the authors propose to use multiple focal planes as input of a deep network. The authors introduce a multi-focal image fusion network, which includes early and late feature fusion, which shows consistent improvements in performance compared to using single focal planes.

**Strengths:**

- The problem tackled is challenging.
- The methodology proposed is sound, and original compared to previous literature.
- Each module of the proposed network is properly validated, with sound contributions to the overall performance.

**Weaknesses:**

- Reproducibility. Although the code is promised to be publicly available, this is not the case for the used dataset. There are few datasets publicly available that could be considered for evaluation of the generalization capabilities of the proposed approach, i.e. [a] and [b], and would ensure better reproducibility and thus encourage a better advancement of the state-of-the-art. Other questions are: 1) the criteria for selecting each microscopic image for each case is not specified, e.g. are they expert-guided selected, or are they based on a particular development stage (e.g. day 2)? 2) Train/validation/Test splits are not specified.
- Methodological related works. The multi-focal network integration resembles related works on multi-modal medical image classification, such as HyperDense-Net [c]), which combines MRI modalities with both early and late feature fusion. Although a different medical image application, it would be beneficial to discuss similarities to multi-modal literature and architectural differences to relevant works in this field.
- Implementation details. I have some doubts regarding the network training. There is a backbone for each focal plane, all of them individual ResNets. Do the features remain frozen during training? Do they share parameters between focal planes?

[a] An annotated human blastocyst dataset to benchmark deep learning architectures for in vitro fertilization, Scientific Data, 2023.

[b] A time-lapse embryo dataset for morphokinetic parameter prediction, Data in Brief, 2022.

[c] HyperDense-Net A hyper-densely connected CNN for multi-modal image segmentation, TMI, 2018.

**Detailed Comments:**

- In my opinion, I feel that the penultimate paragraph of the Introduction “To this end…”, and the Contributions paragraph could be integrated since they share common information.
- In the introduction, when discussing methods that are based on staging classification, authors claim that “they do not directly predict implantation outcomes”. Nevertheless, these methods can correlate implantation outcomes based on elapsed times between stages [b,d, e]. Given that implantation prediction has yet limited accuracy performance in the literature, being such a challenging task, I do not see that a two-step prediction is a motive to disregard those methods, as they might contain complementary information to grading features.

[d] The use of morphokinetics as a predictor of embryo implantation, Human Reproduction, 2011.

[e] Predicting the Success of Blastocyst Implantation from Morphokinetic Parameters Estimated through CNNs and Sum of Absolute Differences, EUSIPCO, 2019.

**Justification Of Final Rating:**

Most of my concerns have been nicely addressed. Nevertheless, the absence of independent testing on additional datasets hardens the reproducibility of the work, and makes me cautious about increasing my score.

**Justification Of The Preliminary Rating:**

The paper's methodological contributions are sound, although there exist some minor weaknesses in reproducibility, lack of references to methodological-related works in medical image analysis such as multi-modality networks, and unclear implementation details.

**Questions To Address In The Rebuttal:**

Please, see Weaknesses.

**Special Issue:**

No

---

> ### Author Response · Authors · 2024-03-15
>
> Thanks for your very thorough review and constructive comments, which are very helpful to improve the quality of our manuscript. Our detailed responses are as follow.
>
> W1:
> 1. We reviewed these two papers. The time-series dataset records multiple stages of embryo development, while the focal plane dataset focuses on a specific stage. Our dataset utilizes multi-focal plane recording to provide a more comprehensive observation of embryo development, hence making it impossible to compare with these methods.
> 2. The shooting time is fixed, i.e., the fifth day of embryo development, and the focus of the focal plane image is determined by experts.
> 3. We employed 5-fold cross-validation and all results are averaged across the five datasets. We pre-divided the dataset using an 80%:20% stratified sampling, randomized five times for training-validation sets.
>
> W2:
> 1. At that time, we also researched the mentioned article. In our experiments, we found that the method of pairwise feature fusion initially increases the number of interactions as modalities increase. For instance, in MRI-CT fusion, only one feature exchange is needed, ensuring that the information of the current modality is not overwritten. However, with an increase in modalities, the proportion of each modality's contribution decreases, leading to an overall increase in redundancy and a decrease in modality independence.
> 2. We also experimented with cross-attention between pairs, but found that the results were not as effective as simply concatenating the images together. We attribute this to the loss of information caused by feature redundancy, ultimately resulting in decreased performance. We will supplement this explanation in the paper.
>
> W3: No, here we need multiple ResNet models with different weights to extract different features separately. The ResNet weights are trainable and not frozen. We will explain this experimental setup in the appendix.
>
> DC1: Thank you for your suggestion! We will consolidate the relevant content as you recommended.
>
> DC2: Here, the purpose is to illustrate the differences between different tasks. While scoring embryos can indeed guide the final prediction results, it is not a comprehensive condition. We hope that the neural network can directly predict the final outcome, allowing the model to spontaneously learn about some areas that we have not pre-defined (areas beyond the stage, ICM and TE).

---

> > ### Comment · Reviewer_eE6q · 2024-03-23
> >
> > I would like to thank the authors for their clarifications. Most of my concerns have been nicely addressed. I will keep my initial recommendation: weak accept. The absence of independent testing on additional datasets hardens the reproducibility of the work, and makes me cautious about increasing my score. I would want to kindly ask the authors to describe in the revised manuscript all additional implementation details raised during the review process but not included in the original submission, and include the discussions raised during the reviewer JMB8 rebuttal and myself regarding the constraints of existing open-access datasets to validate the proposed method. Finally, sharing code implementation of the proposed methods, and if possible, the used dataset, would be largely desirable.

---

> ### Author Response · Authors · 2024-03-25
>
> Regarding the dataset issue, we exerted our utmost effort to locate publicly accessible datasets, unfortunately, we did not come across any datasets, which were not publicly accessible or they were incongruous with our MFIF-Net. We will consider releasing our dataset publicly, and consultation can be conducted to utilize our dataset for research purposes.
>
> Regarding reproducibility: Model implementation details are available on https://github.com/Ch3ngY1/MFIF-Net/tree/master.

---

### Official Review · Reviewer_JMB8 · 2024-03-04

**Confidence:** 4
**Preliminary Rating:** 2
**Final Rating:** 2

**Summary:**

The authors introduce a model named MFIF-Net, designed to predict in vitro fertilization (IVF) blastocyst implantation outcomes. This model employs a deep learning-based approach to fuse multiple focal-plane (FP) images into a single core image. Subsequently, it utilizes both early and late fusion strategies through attention-based modules to extract and refine features for binary classification. The performance of MFIF-Net is evaluated against adaptations of state-of-the-art (SOTA) approaches on their private dataset that comprises 643 images.

**Strengths:**

The paper is well-written, providing clear explanations of its methodologies and findings, which enhances its accessibility to a broad audience.

It introduces a deep learning approach for creating a 'Core' image from multiple FP images, a technique used for the analysis of IVF embryos. The paper also details the use of an attention-based fusion module to integrate features across different focal planes with the Core image, aiming to increase the accuracy of implantation outcome predictions.

To validate their approach, the authors employ a five-fold cross-validation method.

Lastly, the authors plan to make their code publicly available if the paper is accepted, which will contribute to the transparency and reproducibility of their research. This open-science practice is beneficial for fostering further research and collaboration in the field.

**Weaknesses:**

The paper presents an interesting approach, but there are a few areas where improvements could enhance its contribution to the field.

Firstly, the literature review appears to overlook several recent studies. Specifically, such as those found in references 10.1016/j.heliyon.2021.e06298, 10.1109/ACCESS.2021.3053098, 10.3390/s21030863, 10.1155/2023/2426601, and particularly 10.1117/12.2678962, which discusses the use of advanced neural network architectures like the Swin Transformer. Incorporating a discussion on these could provide a more comprehensive comparison and highlight the novelty of the proposed method more effectively.

Moreover, the dataset size utilized in this study is relatively small (643 images) compared to similar works in the domain, such as the one referenced with DOI 10.1109/ACCESS.2021.3053098, which utilized a dataset of approximately 11K images. Expanding the dataset or providing justification for the current size's adequacy in terms of achieving robust and generalizable results would strengthen the paper's validation aspect.

Lastly, the paper does not mention any attempts to test the proposed method on publicly available datasets related to similar tasks. Testing on such datasets could significantly enhance the credibility and applicability of the research findings across different scenarios, contributing to a more solid validation of the proposed approach.

**Detailed Comments:**

In 1. Introduction:

"... embryonic information contained in different FP-images is different, and treating them as equally important may make it difficult to fully exploit the features captured by different focal planes ...": It's important to note that the assertion that all regions within a focal plane are treated equally by deep learning models is not entirely accurate. In fact, these models have the capability to selectively focus on specific regions within each focal plane that are deemed relevant for minimizing the loss function. This allows the models to prioritize certain areas over others during the analysis process. For concrete evidence of this behavior, one can refer to previous studies, such as the one documented in the publication with DOI 10.1109/ACCESS.2021.3053098. This study utilized Grad-CAM visualizations to demonstrate how models discern and utilize different regions of the input images in their decision-making processes.

"... which may have strong correlation with the final result.": Please avoid using phrases like 'which may,' as they can introduce ambiguity. It is essential to adhere strictly to factual information and refrain from speculation. Has the claim been tested and validated with statistical significance? If it has been established in the state of the art, please cite the relevant previous works. If this claim is a finding from your own research, ensure you provide concrete data to support it. Reference the specific paragraph where this evidence can be found to substantiate your claim.

"... , most known fusion methods utilize two modalities...": Please ensure to reference the relevant literature to support your assertions.

"... are relatively easy to fuse.": To enhance the clarity of your argument, could you elaborate on why these elements are easily fusible? Providing additional details will help to understand the underlying reasons for their compatibility and the mechanisms that facilitate their fusion

"... FP images with different key information ...":To better understand your point, could you specify the different types of information you're referring to? Please provide a detailed description of these differences, including how they manifest and their significance. Additionally, it would be beneficial to support your explanation with relevant data to substantiate your observations.

**Justification Of Final Rating:**

I appreciate the authors' efforts in providing clarifications. Nonetheless, upon a detailed examination, I still lean towards a weak rejection of the submission. My decision is anchored in several critical concerns, the most significant of which is the lack of independent testing across diverse datasets. This concern, also raised by other reviewers, brings into question the reproducibility and generalizability of the study's findings.

Another crucial area of critique is methodological shortcomings, particularly the absence of a clear strategy for managing RGB values within a 0 to 1 (or -1 to 1) range during image synthesis via weighted summation of focal plane images. This technique may work for a limited number of focal planes, but it shows scalability limitations with larger datasets, which may include numerous focal planes, potentially affecting the results' integrity.

Furthermore, the paper's discussion on the use of pre-trained models lacks depth. While it mentions using a pre-trained ResNet18 model from ImageNet for the MFIF-Net, it falls short in elaborating on the consideration of other architectures, such as ResNet50, DenseNet, InceptionNet-V1, and EfficientNetV2, in its comparative analysis (refer to Table 1). For scientific precision, it's imperative that the paper clarifies whether these compared architectures were also trained on ImageNet.

Although the authors detail a 5-fold cross-validation in their rebuttal, they do not adequately explain the selection process for the fixed test set, leaving unclear whether it was randomly selected or handpicked.

These points underline my concerns about the study's methodological soundness. I defer to the Area Chair to weigh the issues highlighted by myself and other reviewers (eE6q and RhVt). I encourage the authors to integrate all additional details and discussions from the review and rebuttal phases into a revised manuscript. This should include addressing the challenges of using existing open-access datasets for validating their method. Additionally, making the code and, if possible, the dataset public would significantly improve the transparency and reproducibility of the research, potentially enhancing its acceptance at MIDL or other scientific venues.

**Justification Of The Preliminary Rating:**

My decision to give a weak reject is influenced by critical elements in the manuscript that require further clarification and enhancement for a more robust scientific contribution. A notable concern is the lack of a clear strategy for maintaining the RGB value scale within the 0 to 1 range when generating the core image through weighted summation of focal plane images. The potential for values to exceed this range without a specified clipping process or a dedicated loss function to ensure image soundness raises questions about the methodology's rigor.

Furthermore, the manuscript's discussion on modifying state-of-the-art methods to fit the dataset lacks depth, particularly in specifying the levels at which these modifications were applied. This lack of detail hampers the understanding of how the proposed approach diverges from or improves upon existing techniques.

The use of pre-trained backbones, such as ResNet18 from ImageNet, is mentioned but not explored in detail, especially concerning whether similar pre-trained weights for other models were employed comparably in the analysis. Explicitly stating the use of pre-trained weights is crucial for assessing the scientific validity of the comparative analysis presented.

Lastly, the absence of independent public datasets in the comparison limits the ability to benchmark the proposed approach against current state-of-the-art methods comprehensively. Including such datasets would significantly enhance the evaluation of the model's performance, providing a more grounded understanding of its efficacy and applicability in broader contexts. These areas of concern are central to my decision, highlighting the need for more detailed information and methodological rigor to fully substantiate the claims and contributions of the manuscript.

**Questions To Address In The Rebuttal:**

In 2.1. Core Image Generator (CI-Gen):

"... are usually fused with each other, but this fusion strategy is not suitable for fusing three modalities.": Could you please provide further clarification on this matter?

"... the core image $I_{core}$ is generated by weighted summation of the three FP-images ...": In reviewing the manuscript, I noticed there is no mentioned strategy for maintaining the scale of the resulting image, specifically keeping RGB values within the 0 to 1 range. When summing the weighted images, it appears possible for the values to exceed 1. Is there a clipping process applied to address this issue? Could you please provide more details on how the scale of the resulting image is managed?


In 3. Experimental Results:

"...  five-fold cross-validation ...": I'm interested in understanding the methodology behind the selection of the fixed testing set described in your study. Could you provide more detailed information on how this set was chosen? Additionally, I'm curious about the decision not to utilize a five-fold cross-testing approach, which would involve changing the test set with each iteration. What were the reasons behind opting for a fixed test set instead?


In Comparison to State-of-the-Art Methods:

"We modify known state-of-the-art (SOTA) methods to fit our dataset...": Could you provide further details on the specific level at which modifications were applied? Elaborating on this will significantly enhance our comprehension of the adjustments made.

Additionally, the document does not discuss the use of pre-trained backbones in depth. You mentioned employing a pre-trained ResNet18 backbone from ImageNet for your MFIF-Net model, but it remains unclear whether comparable pre-trained weights for ResNet50, DenseNet, InceptionNet-V1, and EfficientNetV2 were utilized in your comparative analysis. If they were used, it would be crucial to explicitly state this to ensure the scientific validity of the comparisons made.

Lastly, I recommend including independent public datasets in your comparison to benchmark your approach against current state-of-the-art methods. This would provide a more comprehensive evaluation of your model's performance.

In 3.2. Ablation Study:

For a more comprehensive analysis, it would be beneficial to incorporate a z-max projection as an additional test. This process involves using image processing to determine the maximum value per pixel across the Stage, ICM, and TE, in order to create the Core image. Implementing this technique will serve as a valuable comparison, offering clear evidence of the effectiveness and utility of the deep learning (DL)-based weighting approach. Such an inclusion would greatly enhance the robustness of your findings by providing a direct contrast between traditional image processing methods and the proposed DL methodology.

Finally, will the dataset used be made publicly available to the scientific community?

---

> ### Author Response · Authors · 2024-03-16
>
> Thank you for your review comments. In order to address the confusion you raised, we will detail our supplementary experiments and paper explanations below.
>
> W1: Thank you for your suggestion! We have reviewed the overlooked references and will add them into related work. Additionally, we have included the experimental results for 10.3390/s21030863: ACC=62.0, F1=61.7, AUC=56.1, SEN=53.2, PPV=62.6, NPV=61.5. Due to the differing data formats (more focal planes, sequential data) of other methods, we did not compare with them.
>
> W2: The dataset used in DOI 10.1109/ACCESS.2021.3053098 contains 1K samples, each sample has 11 focal plane images. Our dataset consists of 600+ samples, each with 3 focal plane images. Since predictions are sample-specific, the gap in sample sizes between our dataset and theirs isn't significant. Due to the inability of single images to depict the overall information of samples, we did not employ the training-prediction method using single images.
> Finally, to mitigate the impact of samples on results, we employed five-fold cross-validation to average out fluctuations across different data splits, and our results are the average of multiple folds.
>
> W3: Thank you for raising this important point. We acknowledge that the ethical considerations surrounding embryo data and the unique nature of multi-focal plane photography posed significant challenges in finding suitable datasets for evaluation. Despite our best efforts to identify comparable data, we were unable to locate datasets that closely matched the specific characteristics and structure of our study. But, we believe that our study still provides valuable insights in the context of embryo assessment using multi-focal plane imaging. In our future work, we will focus on establishing standardized datasets and protocols for embryo imaging, taking into account the ethical concerns and the need for diverse, representative data. Collaboration among research institutions and ethical oversight committees could facilitate the creation of such datasets, enabling more comprehensive evaluations and comparisons of different methodologies in this field.
>
> DC1:
> The phrase "treating them as equally important" refers to the need for equivalent treatment of different focal plane images during processing, rather than assigning a primary modality (such as visual information) and treating others (such as audio and thermal images) as auxiliary modalities, as done in some multi-modal fusion methods. Our heatmaps also indicate that the model can automatically focus on different regions of different focal plane images. We will release heatmap visualizations in subsequent versions. We apologize for any confusion caused by our description.
>
> DC2:
> We will revise this section. The quality of ICM, TE, etc. has literature support, which is mainly determined by the Gardner blastocyst grading system, which based on the following criteria:
>
> 1.	Degree of blastocyst expansion: Assessing the degree of blastocyst expansion, categorized into five grades ranging from early to full expansion.
>
> 2.	Quality of inner cell mass (ICM): Evaluating the appearance and organization of cells within the inner cell mass.
>
> 3.	Quality of trophectoderm (TE): Evaluating the appearance and cell density of the trophectoderm.
>
> 4.	Embryo uniformity: Assessing the consistency and cohesion between the inner cell mass and trophectoderm.
>
>
> DC3&DC4:
> We will add relevant citations. Here, we primarily investigated Fusion Transformer$^{[1]}$, Attention Bottlenecks$^{[2]}$, and HyperDense-Net$^{[3]}$, which are two-modalities fusion methods. The phrase "easy to fuse" in the text refers to the ease of transferring between different types of two-modal datasets, but it is challenging to adapt to more modalities. Therefore, pair-wise feature interaction exhibits a complexity of $N^2$ as the number of modalities increases.
>
> [1]Multi-Modal Fusion Transformer for End-to-End Autonomous Driving
>
> [2]Attention Bottlenecks for Multimodal Fusion
>
> [3]HyperDense-Net A hyper-densely connected CNN for multi-modal image segmentation
>
> DC5:
> This is also related to the Gardner grading system mentioned in DC2. The ICM mainly focuses on the cell number of the inner cell mass, while TE pays attention to the cell number and density of the TE layer. The stage mainly focuses on the morphology of the ZP. We will provide detailed supplementary explanations for these descriptions.

---

> > ### Author Response · Authors · 2024-03-17
> >
> > Q1: In multi-modality, we found that pairwise fusion of the three focal plane images (e.g., cross-attention) yielded poor results, even worse than simply concatenating the three images together. In other works, fusion typically involves combining N-1 modalities into one primary modality, but our three focal planes are equivalent, rendering such approaches inapplicable. Ultimately, we believe that multi-modal algorithms are not well-suited for three images.
> >
> > Q2: The fusion ratio is determined by the model's activation values, hence not fixed. We didn't use softmax because some regions have clear focus across multiple focal plane images, making it challenging to explicitly represent their weights after softmax normalization. Conversely, in regions with unclear focus, we aimed to suppress the entire region, hence avoiding softmax normalization.
> >
> > Q3: We employed the widely used five-fold cross-validation to ensure fairness across methods. Initially, we created five training-validation sets, each containing 80% positive and negative samples for stratified sampling. Due to the limited overall training data, we used five-fold cross-validation to mitigate fluctuations resulting from data split averaging. Based on our prior experience, we utilized the train-validation split for dataset division.
> >
> > Q4: Due to the scarcity of multi-focal plane classification methods, we converted the backbones of other embryo-related methods into late fusion and early fusion as additional baselines.
> >
> > Q5: We compared the performance of ResNet-50 and ResNet-18 backbones in our experiments and found minimal overall differences, leading us to use ResNet-18 as the backbone. While other backbones could be selected, this isn't our paper's focus. Based on our experimental experience, we believe that ResNet can demonstrate the benefits of our innovation, hence we didn't employ other methods.
> >
> > Q6: Thank you for your suggestion. However, in our previous research, we didn't encounter scenarios with fixed quantities of focal plane data. Although a method mentioned multi-focal planes, the number of focal planes ranged from 3 to over 100, making replication unfeasible.
> >
> > Q7: Thank you for your suggestion. We will incorporate z-max projection into our experiments. The experimental results are as follows: ACC=58.1, F1=54.9, AUC=54.7, SEN=30.6, PPV=63.7, NPV=56.2. These results indicate that z-max performs poorly in embryo image fusion. This is because z-max is often used in 3D images like CT, where brightness variations represent corresponding physical properties. However, in embryo images, these pixel values don't convey any particular information.
> >
> > Q8: We will make the codes publicly available upon publication of the manuscript. Anonymized EEG datasets may be provided upon reasonable request, in accordance with the cooperating hospital's research data access policies.

---

> > > ### Comment · Reviewer_JMB8 · 2024-03-23
> > >
> > > I appreciate the authors' efforts in providing clarifications. Nonetheless, upon a detailed examination, I still lean towards a weak rejection of the submission. My decision is anchored in several critical concerns, the most significant of which is the lack of independent testing across diverse datasets. This concern, also raised by other reviewers, brings into question the reproducibility and generalizability of the study's findings.
> > >
> > > Another crucial area of critique is methodological shortcomings, particularly the absence of a clear strategy for managing RGB values within a 0 to 1 (or -1 to 1) range during image synthesis via weighted summation of focal plane images. This technique may work for a limited number of focal planes, but it shows scalability limitations with larger datasets, which may include numerous focal planes, potentially affecting the results' integrity.
> > >
> > > Furthermore, the paper's discussion on the use of pre-trained models lacks depth. While it mentions using a pre-trained ResNet18 model from ImageNet for the MFIF-Net, it falls short in elaborating on the consideration of other architectures, such as ResNet50, DenseNet, InceptionNet-V1, and EfficientNetV2, in its comparative analysis (refer to Table 1). For scientific precision, it's imperative that the paper clarifies whether these compared architectures were also trained on ImageNet.
> > >
> > > Although the authors detail a 5-fold cross-validation in their rebuttal, they do not adequately explain the selection process for the fixed test set, leaving unclear whether it was randomly selected or handpicked.
> > >
> > > These points underline my concerns about the study's methodological soundness. I defer to the Area Chair to weigh the issues highlighted by myself and other reviewers (eE6q and RhVt). I encourage the authors to integrate all additional details and discussions from the review and rebuttal phases into a revised manuscript. This should include addressing the challenges of using existing open-access datasets for validating their method. Additionally, making the code and, if possible, the dataset public would significantly improve the transparency and reproducibility of the research, potentially enhancing its acceptance at MIDL or other scientific venues.

---

> > > > ### Author Response · Authors · 2024-03-25
> > > >
> > > > We appreciate for your timely feedback and provide further clarifications about your remain concerns as follows:
> > > >
> > > > Regarding the issue with the dataset, we made our extensive efforts to find publicly available datasets that align with our research objectives, but unfortunately, the available datasets were either not directly comparable to our study or lacked the necessary characteristics for our methodology. Also, upon careful examination of those datasets in the suggested literature, we found that these datasets were either not publicly available or cannot be applied to our method as we have mentioned. Finally, interested researchers can contact us to utilize our dataset for their own research purposes.
> > > >
> > > > Regarding the RGB normalization issue, as mentioned in our previous response, the reason we didn't use softmax is because we wanted to maintain activations for multiple focal planes in certain areas, while keeping them invalid for multiple focal planes in blank areas, hence we didn't employ softmax.
> > > >
> > > > Regarding the concern of methodological shortcomings. For the RGB values, our aim is to enhance the final classification performance. The fused images only need to be "model-friendly" for this purpose. For numerous focal planes, our method can be naturally extended by incorporating a preprocessing step as mentioned in [1], where we initially filtered images using similarity computation. Through this way, we can significantly reduce redundancy and computational complexity.
> > > >
> > > > Regarding the pre-trained model description issue, we apologize for the confusion. All the deep models used in our paper were initialized with ImageNet1K pretrained weights. Our experiments have demonstrated that models initialized with pretrained weights exhibit stronger performance. If the paper is accepted, we will include experimental results for additional backbones in the final version.
> > > >
> > > > Regarding cross-validation, we have mentioned that the five-fold cross-validation involved randomly stratified sampling of positive and negative samples five times to obtain five fixed datasets. For each dataset, the model was trained on the training set and validated on the validation set, with the final result being the average over the five datasets [2].
> > > >
> > > > Regarding reproducibility: Model implementation details are available on https://github.com/Ch3ngY1/MFIF-Net/tree/master.
> > > >
> > > > [1] Deep Learning for Human Embryo Classification at the Cleavage Stage (Day 3)
> > > >
> > > > [2] https://en.wikipedia.org/wiki/Cross-validation_(statistics) # Exhaustive cross-validation

---

### Official Review · Reviewer_RhVt · 2024-03-05

**Confidence:** 4
**Preliminary Rating:** 4
**Recommendation:** Poster
**Final Rating:** 4

**Summary:**

This study focuses on the selection of high-quality embryos and predicting implantation outcomes in IVF treatments to minimize the occurrences of multiple pregnancies (i.e., single-embryo transfer). Typically, single images or focal planes (FPs) are used in such assessments, which involve reviewing various stages of blastocytes. However, not only is this time intensive for clinicians, but these single images are not representative of the 3D nature of the embryo. Ideally, multiple focal-planes (FPs) need to be analyzed. Previous work in this space has utilized three FPs. However, the limitation of these studies was that each plane in the assessment was treated equally (i.e., equal weighting). However, this weighting scheme would not work, given each plane includes different key information. Thus, the authors created MFIF-Net, in which three FPs are fed into their pipeline, and implantation outcomes are predicted by looking at key features across the FPs. The pipeline consists of a CI-Gen (which fuses the three images to generate a clear core image) and KFFNet (that captures key features by a fusion model).

**Strengths:**

-	The use of multi-focus fusion makes sense here. Each stage-based single FP illustrates a region of interest it focuses upon, with the remainder of the image out of focus. Thus, the fusion of these single FPs into one “clear” image is effective (i.e., one in which all regions are clearly visible, and no region is out of focus). Also, the process involves assessing the feature extraction against the four modalities (i.e., the combined core, and the three original images of stage, ICM, and TE) and using feature fusion to create a single feature file (i.e., identifying key features). The pipeline addresses the needs of the clinical task set out.

**Weaknesses:**

The following suggestions will assist in improving the manuscript.

-	Some grammatical errors and typographical errors. For example, in the Introduction, full names for ICM and TE are stated twice. In Section 3.1., they provide an acronym for state-of-the-art but then write the full definition a few times.

-	Odd placement of Figure 1 (i.e., before introduction).

-	In Figure 1, ZP is highlighted but it is very small, so I almost missed it. Consider stacking the images (a) and (b) onto one another and make the images larger. Also, clearer indications on which regions correlate to ZP, ICM, and TE areas.

-	A few additional words to describe why it is challenging to capture distinct features from single images would help frame the problem better for the reader.

-	For Contributions, while the use of this pipeline for embryo implantation appears to be novel, prior work in using Multi-Focus Fusion for embryo assessment needs to be acknowledged (https://doi.org/10.3390/s21030863). The authors provide a comparison for other state-of-the-art, but this publication should also be considered (at least for the fusion of the FPs into one clear image). Also acknowledge other publications that do not use fusion modeling, but have instead used time-lapse image sequences for grading (this was not included in the references - https://doi.org/10.1371/journal.pone.0262661).

-	The authors state the dataset was “ethically approved”, but a quick line regarding the approving institution along with the ethics approval number would be great.

-	The authors do miss a couple of smaller details, such as the image alignment used to account for embryo movement between FPs.

-	It would be nice to see qualitative results in addition to quantitative results to validate the efficiency of the pipeline. For example, what does the final clear image look like? Are there any anomalies that need to be accounted for?

-	The authors provide metrics such as accuracy, sensitivity, etc. when comparing their work with state-of-the-art. However, what about image quality metrics, such as PSNR, particularly to assess the quality of the fusion of the images? Particularly given the pipeline heavily relies on the quality as a result of the fusion process. You can either find one image that is selected as the reference for good quality to run quality metrics across your entire dataset or use a no-reference metric such as NIQE.

**Detailed Comments:**

Overall, good paper, but there are some key considerations for the authors to revisit and improve.

**Justification Of Final Rating:**

I maintain my original score of weak accept. The authors have responded to the queries set out to them efficiently. The weak acceptance is contingent on the authors making the modifications they set out in their responses.

**Justification Of The Preliminary Rating:**

I am convinced by the work and its novelty in the application. However, I feel the manuscript overall needs improvement. This includes providing qualitative alongside quantitative results, addressing issues such as the assessment of the quality of images produced (given the rest of the pipeline is contingent on it), note key previous works that have utilized similar methods (and therefore would be better suited for comparison), etc. These will assist in ensuring the effectiveness of the proposed work.

**Questions To Address In The Rebuttal:**

I would particularly be interested in seeing the qualitative results, and, furthermore, additional comparison with the publication that is close in its premise and methods to the proposed work.

**Special Issue:**

No

---

> ### Author Response · Authors · 2024-03-15
>
> Thanks for your very thorough review and constructive comments, which are very helpful to improve the quality of our manuscript. We provide the   detailed responses and the new result of key previous work below.
>
> W1: We will very carefully read the entire text and correct any typographical errors.
>
> W2: We will adjust the position of the image and place it on the third page, within the introduction section.
>
> W3: We will remake the image to ensure clarity.
>
> W4: Thanks for your suggestion. Due to the focus of a single image on a specific area, extracting features from the entire image can result in excessive redundancy. This leads to a lower proportion of weight assigned to relevant information, thereby affecting classification performance. We appreciate your valuable feedback and will clarify this if accepted.
>
> W5:
>
> 1.	The literature (https://doi.org/10.3390/s21030863) indeed presents a multi-focus plane fusion method, which we will regard it as a comparative baseline as mentioned in the final version. Following your suggestion, we implemented this method and found the result achieved ACC=62.0, F1=61.7, AUC=56.1, SEN=53.2, PPV=62.6, NPV=61.5. From the results, it's evident that the performance of this method is indeed superior to both late fusion and early fusion. We attribute this to the more complex image fusion process provided by UNet. However, due to the characteristics of early fusion, which lack interaction between different focal plane images during feature extraction, the overall performance is slightly inferior to our MFIF-Net.
>
> 2.	However, the time-lapse photography method (https://doi.org/10.1371/journal.pone.0262661) is not applicable in this paper because our dataset only considers single stages (D5), which is not suitable for a sequence classification method. Therefore, this paper is not mentioned, but we will discuss it in the related work section.
>
>
> W6: Very thanks for your remind. Actually, we have obtained the ehical approval for scientic research, specifically Sichuan Jinxin Xinan Women & Children Hospital (2019) Reproductive Ethics Approval No. (002). This information will be added in the revised version.
>
> W7: This part of the process occurs during the image preprocessing stage. We utilize the Enhanced Correlation Coefficient (ECC) algorithm to compute the transformation matrix between two images, and then use this matrix to align the second image. If the paper is accepted, we will add this in the subsection Appendix A and also release the preprocessing code publicly.
>
> W8: The mixed image is optimized for neural networks, so its final effect may not be easily understandable to humans. However, its effectiveness can be reflected through heatmaps, which we will include heatmap images in the final version.
>
> W9: Since we lack ground truth for mixed images, it's not feasible to calculate metrics like PSNR. However, heatmap analysis reveals that the model indeed focuses on the corresponding areas of each focal plane image. Additionally, it is important to note that the primary motivation behind fusing multiple focal plane images is not to enhance the visual quality of the fused image itself, but rather to leverage the complementary information present in each image to boost the model's predictive capabilities.

---

> > ### Comment · Reviewer_RhVt · 2024-03-28
> >
> > I thank the authors for their time and effort in responding to all the reviewer comments. This includes providing clarification on the lack of availability of open source datasets for external validation, making some of their code publicly available for repeatability and reproducibility, and running additional comparative analyses to other state-of-the-art mentioned in the original review.
> >
> > While the authors have mostly addressed my original concerns, the one thing that remains outstanding is the image quality checks. In their response, the authors state they cannot carry out analyses such as PSNR in the absence of a ground truth, which is correct. However, in my original review, I also mentioned no-reference measures. At the time of my review, I suggested NIQE, which uses natural scene statistics rather than a reference image for its calculations. However, I am happy to change this to BRISQUE, as it is already easily accessible through PyTorch and would not take the authors that long to implement and report.
> >
> > Thus, I maintain my original score of weak accept, granted that the authors do make the revisions they state in their responses, including the addition of comparative results with Raudonis et al. (2021; https://doi.org/10.3390/s21030863), run additional analyses using BRISQUE to assess the quality of their fused images, and in line with the recommendations of the other reviewers, release all code publicly.

---

### Meta-Review · Area_Chair_uNV5 · 2024-03-30

**Recommendation:** Accept (Poster)
**Confidence:** 4

**Metareview:**

The majority of the reviewers render a positive final rating.

The work details a study focused on enhancing embryo selection and predicting implantation outcomes in IVF treatments through the development of the MFIF-Net model, offering several noteworthy contributions and areas for improvement. MFIF-Net analyzes multiple focal-plane images of embryos, and demonstrates a sophisticated attempt to address the three-dimensional complexity of embryos.  The paper provides clear explanations of methodologies and findings, enhancing accessibility to a broad audience. This clarity is particularly crucial in a field where interdisciplinary knowledge is essential.

[Pros]
Methodological rigor and originality in addressing a complex problem. Clear presentation and thorough validation of the proposed model. Potential for significant clinical impact in IVF treatments.

[Cons]
Some clarity issues in the presentation, particularly regarding the literature review and methodological comparisons. Limited dataset and lack of independent external validation limit the generalizability of the findings. The reproducibility of the work could be improved by providing more detailed implementation details and considering the public release of the dataset.

---

### Decision · Program_Chairs · 2024-04-05

Accept (Poster)